# Machine Learning-Driven Classification of Urease Inhibitors Leveraging Physicochemical Properties as Effective Filter Criteria

**DOI:** 10.3390/ijms25084303

**Published:** 2024-04-13

**Authors:** Natalia Morales, Elizabeth Valdés-Muñoz, Jaime González, Paulina Valenzuela-Hormazábal, Jonathan M. Palma, Christian Galarza, Ángel Catagua-González, Osvaldo Yáñez, Alfredo Pereira, Daniel Bustos

**Affiliations:** 1Magíster en Ciencias de la Computación, Universidad Católica del Maule, Talca 3460000, Chile; nmoralesr@ucm.cl (N.M.); jaime.gonzalez@alu.ucm.cl (J.G.); 2Doctorado en Biotecnología Traslacional, Centro de Biotecnología de los Recursos Naturales, Universidad Católica del Maule, Talca 3480094, Chile; elizabeth.valdes@alu.ucm.cl; 3Departamento de Farmacología, Facultad de Ciencias Biológicas, Universidad de Concepción, Concepción 4030000, Chile; paulinvalenzuela@udec.cl; 4Facultad de Ingeniería, Universidad de Talca, Curicó 3344158, Chile; jonathan.palma@utalca.cl; 5Departamento de Matemáticas, Facultad de Ciencias Naturales y Matemáticas, Escuela Superior Politécnica del Litoral, Guayaquil EC090903, Ecuador; chedgala@espol.edu.ec (C.G.); anglucat@espol.edu.ec (Á.C.-G.); 6Núcleo de Investigación en Data Science, Facultad de Ingeniería y Negocios, Universidad de las Américas, Santiago 7500000, Chile; oyanez@udla.cl; 7Facultad de Ingeniería, Arquitectura y Diseño, Universidad San Sebastián, Bellavista 7, Santiago 8420524, Chile; 8Laboratorio de Bioinformática y Química Computacional, Departamento de Medicina Traslacional, Facultad de Medicina, Universidad Católica del Maule, Talca 3480094, Chile

**Keywords:** urease inhibitors, cheminformatics, machine learning, predictive modeling, bioactivity prediction, classification models

## Abstract

Urease, a pivotal enzyme in nitrogen metabolism, plays a crucial role in various microorganisms, including the pathogenic *Helicobacter pylori*. Inhibiting urease activity offers a promising approach to combating infections and associated ailments, such as chronic kidney diseases and gastric cancer. However, identifying potent urease inhibitors remains challenging due to resistance issues that hinder traditional approaches. Recently, machine learning (ML)-based models have demonstrated the ability to predict the bioactivity of molecules rapidly and effectively. In this study, we present ML models designed to predict urease inhibitors by leveraging essential physicochemical properties. The methodological approach involved constructing a dataset of urease inhibitors through an extensive literature search. Subsequently, these inhibitors were characterized based on physicochemical properties calculations. An exploratory data analysis was then conducted to identify and analyze critical features. Ultimately, 252 classification models were trained, utilizing a combination of seven ML algorithms, three attribute selection methods, and six different strategies for categorizing inhibitory activity. The investigation unveiled discernible trends distinguishing urease inhibitors from non-inhibitors. This differentiation enabled the identification of essential features that are crucial for precise classification. Through a comprehensive comparison of ML algorithms, tree-based methods like random forest, decision tree, and XGBoost exhibited superior performance. Additionally, incorporating the “chemical family type” attribute significantly enhanced model accuracy. Strategies involving a gray-zone categorization demonstrated marked improvements in predictive precision. This research underscores the transformative potential of ML in predicting urease inhibitors. The meticulous methodology outlined herein offers actionable insights for developing robust predictive models within biochemical systems.

## 1. Introduction

### 1.1. Urease Enzyme and Its Implications in the Human Context

The urease enzyme is a key element in nitrogen (N) metabolism in bacteria, fungi, algae, and plants, hydrolyzing urea (carbamide) over 10^14^-fold the conversion rate to ammonia and CO_2_ [1,2]. One of the most conflictive ureolytic bacteria for humans is *Helicobacter pylori* (H.p.), which is very well adapted to survive in a wide range of environments by secreting a high number of urease enzymes, even in acidic environments such as the stomach [3,4]. It is estimated that over 50% of the world population is already infected by H.p. [5,6]. Once H.p. colonizes the host, this Gram-negative bacterium increases the risk for peptic ulcers [7,8], chronic kidney diseases [9], idiopathic thrombocytopenic purpura [10,11], iron deficiency anemia [12], and gastric cancer (GC) such as gastric adenocarcinoma [13,14] and MALT (mucosa-associated lymphoid tissue) lymphoma [15,16,17]. H.p. is the only bacterium classified in Group I of carcinogens to humans by the International Agency for Research on Cancer, where 89% of all GC is related to H.p. infections [18]. Until 2020, GC was one of the deadliest types of cancer worldwide [19]. In this sense, it has been evidenced that H.p. eradication reduces the mortality rate caused by GC [20]. The current treatment proposed to eradicate H.p. infections is to combine broadband antibiotics (amoxicillin, metronidazole, and clarithromycin) with a proton-pump inhibitor or with bismuth-containing compounds [21,22,23]. Nowadays, these therapies have become unfeasible due to both the alarming resistance of H.p. to antibiotics worldwide and the side effects (nausea, diarrhea, headache, angioedema, and microflora disorders) produced by antibiotics. New therapeutical strategies based on the use of urease inhibitors (UIs) have been proposed to treat infections of urease-dependent microorganisms such as H.p since inhibiting the ureolytic faculty in H.p. causes this bacterium to become unable to cause infections in animal models [24]. However, available classic UIs (sulfhydryl compounds, amides and esters of phosphoric acid, hydroxamic acid derivatives, and imidazoles) are toxic for humans, which precludes their clinical uses [25].

Many efforts are being focused on searching for potent UIs [26]. Natural sources provide an immeasurable number of organic compounds with anti-urease activity, and these compounds could be used as a starting point to design stronger inhibitors. At present, there exist compounds reported with anti-urease activity such as polyphenols, flavonoids [27,28,29], alkaloids [30], triazole [31,32,33,34,35] thiadiazole [24,30,31,36], and coumarins [24,37,38,39,40,41]. Despite the existence of a significant repertoire of urease inhibitors, there are compelling reasons to continue the pursuit of novel inhibitors. This drive is fueled by the limitations of conventional methods and the potential advantages offered by unconventional techniques, such as ML. These ML models can provide insights into the structure–activity relationships of urease inhibitors, aiding in the rational design of novel compounds while reducing time and resources.

### 1.2. Machine Learning in the New Era of Computer-Aided Drug Discovery

It is projected that bringing a drug from its initial stage to market may take up about a decade and incur expenses exceeding USD 2.8 billion [42]. The early stage of drug discovery, also named computer-aided drug discovery (CADD), has rapidly emerged along with the development of structural biology and the computational power of new hardware [43]. CADD is a collection of diverse computational techniques and resources, comprising compound databases, molecular simulations, structure- and ligand-based virtual screenings (VS), hit and lead optimization, quantitative structure–activity relationship (QSAR), among many others. The integration of various ML algorithms into the CADD process has greatly benefited pharmaceutical companies and academic research as ML provides them with innovative and efficient ways in every stage of the CADD process [44,45] and other branches of chemistry [46,47,48]. In ML, there exist two primary categories of algorithms: supervised learning and unsupervised learning. The first is responsible for learning from labeled training samples to determine the labels of new samples, whereas the latter is responsible for identifying patterns within an unlabeled dataset. Typically, before pattern recognition, high-dimensional data are transformed into a lower dimension via unsupervised learning algorithms to increase efficiency [49]. Through the use of ML algorithms, various models have been created which allow for a more precise understanding of the biochemical and physical–chemical characteristics of candidate compounds in VS protocols, thus allowing a reduction in false positives and false negatives [50]. In this sense, Liu and colleagues utilized a support vector machine (SVM) algorithm to construct various classification models for 85 specific cyclooxygenase-2 inhibitors featuring the 1,5-diarylimidazoles scaffold. The optimal classification models show accuracies of 91.2% and 88.2% for the training and test sets, respectively [51]. Kumar and Patra employed known catechol O-methyltransferase inhibitors as input to discover new inhibitors by combining ML regression and sampling molecular dynamics methods. Their models achieved an R^2^ of over 0.7 for both training and test datasets [52]. In order to deal with the imbalance of inhibitor and non-inhibitor classes, Tinivella et al. employed a flexible thresholding strategy on a set of modulators deposited in ChEMBL of two human carbonic anhydrase isoforms through an ML protocol [53]. In the urease context, to our best knowledge, only two studies have employed ML techniques for the development of new inhibitors. Mermer et al. [54] uses regression and classification models to identify novel thiazole derivatives with a balanced accuracy of 78% and an R^2^ between 0.2 and 0.7. Aniceto et al. [55] discover new inhibitors of *jack bean* urease by using three ML algorithms with 81% precision in its best model.

The main goal of this research is to construct robust and accurate ML models capable of predicting the activity of urease inhibitors in *Hp* (see schematic workflow in Appendix A). Simultaneously, we conduct a thorough exploratory data analysis to identify pertinent features crucial for predicting the behavior of urease inhibitors. Subsequently, our objective is to investigate diverse strategies for categorizing bioactivity and identify suitable ML algorithms to determine the most effective approach for model development. The study encompasses the standard procedures in ML protocols, which include data collection, data preprocessing, exploratory data analysis (EDA), data partitioning, and the learning stages (model selection, training, hyperparameter tuning, and evaluation).

## 2. Results and Discussion

### 2.1. Exploratory Data Analysis

During the analysis of this dataset, we investigated 677 different compounds and 207 variables, with a specific focus on the response variable IC_50_. Opting to utilize pIC_50_ instead of IC_50_ proves beneficial due to the notable variation in concentration ranges exhibited by the latter (ranging from 0.009 µM to 1000 µM within our dataset). IC_50_ values frequently encompass a wide span of orders of magnitude, consequently giving rise to challenges in direct comparison and effective visualization. In contrast, pIC_50_ offers a more concise representation achieved by applying the negative logarithm to IC_50_ values, which are then standardized to a consistent concentration level. This transformation serves a dual purpose: not only does it normalize the data, but it also enhances the comprehension of compound potency across a wider array of concentrations. In Appendix A, the distribution of bioactivity is illustrated, delineating nUIs, UIs, and the intermediate gray-zone compounds generated by the 5–50 µM cutoff.

To identify potential inhibitor features, a correlation analysis is performed with respect to pIC_50_, which substantially increases correlations compared to IC_50_. Figure 1 displays a heatmap of the correlation matrix between pIC_50_ and the 16 features with |ρ| > 0.4. Larger circle diameters indicate stronger correlations. WTPT.4, TPSA, and RPSA exhibit moderate and negative correlations with pIC_50_. These attributes pertain to topological (WTPT.4) and electronic (TPSA and RPSA) characteristics. The initial one, WTPT.4, signifies the molecular branching in each molecule originating from oxygen atoms. On the other hand, TPSA and RPSA correspond to diverse measurements of the solvent-accessible surface area (SASA). Specifically, TPSA represents the sum of the SASA of atoms with an absolute value of partial charges ≥ 0.2, while RPSA is calculated as the ratio of TPSA to the total SASA. Additionality, positive relationships are observed with khs.dsN (Kier–Hall E-state descriptor), C1SP2 (carbon atoms with hybridization Sp2), SCH.6 (Kier and Hall Chi chain index), WTPT.5 (molecular branching starting from the nitrogen), MDEN.22, and MDEN.23 (molecular distance edge descriptors). Significant and strong positive correlations among the independent variables are evident, such as TPSA and RPSA (0.92), FNSA.3 and RHSA (0.94), khs.aasN and MDEN.23 (0.95), as well as RnRings5 and SCH.5 (0.99), among several others. These correlations are unsurprising, given that these descriptors belong to the same categories and are closely interconnected. Given that none of the selected features exhibit a high correlation with the response variable pIC_50_, it suggests a lack of strong linear association between the chemical properties of the compounds and their inhibitory activity against urease. Consequently, the molecular descriptors captured by these variables may not directly or simply correspond to the bioactivity measured by pIC_50_. This circumstance prompts the exploration of supervised machine learning models as they can adeptly capture complex and non-linear relationships between the features and pIC_50_ by discerning non-trivial chemical patterns within our dataset.

The central tendency (median) and variability (interquartile range) measures were studied for the UI and nUI groups with the most correlated features. The boxplots in Figure 2 show clear differences in the 16 relevant features between UI and nUI compounds. The median and variability differ between the two groups, and outliers are present in most features. Features with lower median values for inhibitor compounds include WTPT.4, TPSA, and RPSA, while the remaining features have higher median values allowing separate both classes. Furthermore, WTPT.4, TPSA, and RPSA not only have lower median values for the inhibitor group but also lower variability, resulting in more consistent values and lower uncertainty. Finally, all 16 features show statistically significant mean differences between the two groups based on the *p*-value of a joint Wilcoxon rank sum test.

Figure 3 presents a two-dimensional representation of the data using a PCA. The direction of each arrow represents the direction and magnitude of the maximum variability in the data in the chemical feature space. Each arrow, also known as an eigenvector, indicates the direction in which the data have the greatest variation: the larger the size, the greater the variation. The ellipsoids (red for nUIs and blue for UIs) represent the spread of the data for each class in the feature space reduced by the method. Each ellipsoid describes the cloud of points of a class in the lower-dimensional feature space generated by the PCA. The shape and size of the ellipsoids indicate the variability in and distribution of the data for each class. It is revealed from this visualization that certain features, such as khs.aaNH, SCH.5, SCH.6, C1SP2, VCH.6, and nRings5, are associated with UIs, while TPSA, WTPT.4, and RPSA values are associated with nUIs. The intersection between the ellipsoids represents the region where the two classes have an overlap in the reduced feature space. This overlap indicates that there are data instances that share similar characteristics between both classes. Therefore, the intersection between the ellipsoids may contain points that are difficult to definitively classify as belonging to a specific class, suggesting the presence of instances with ambiguous or intermediate bioactivity. It is mainly on these that we hope that later ML methods will allow them to be classified correctly. Moreover, we conduct a t-SNE analysis, as showcased in Appendix A. t-SNE reveals a separation between UI and nUI classes where compounds situated within the gray zone exhibit overlapping representations in both clusters (UIs and nUIs). This highlights, together with the results of the PCA, the essential requirement to explore more intricate bioactivity categorization approaches, with the aim of precisely distinguishing which compounds can be appropriately classified as UIs. The exploratory analysis conducted underscores the presence of a diverse range of features that facilitate clear distinctions both within individual classes and among different groups. This forms a robust basis for crafting ML classification models, which have the potential to unveil complex and less obvious relationships during the preliminary stages of EDA.

### 2.2. Machine Learning Models

In the present study, an extensive methodology was employed to discover high-performance models for the prediction of urease inhibitors. Seven ML algorithms, random forest (RF), support vector machine (SVM), decision tree (DT), eXtreme Gradient Boosting (XGB), k-nearest neighbor (KNN), naive Bayes (NB), and logistic regression, (LR) and three attribute selection methods, Boruta, XGB, and nFS (non-feature selection), were compared in conjunction with six different strategies for categorizing the bioactivity of the inhibitors, resulting in a total of 126 models. Furthermore, each model was trained twice, considering the attribute “chemical family type” and excluding it. In summary, 252 distinct models were trained.

Figure 4 displays a comparison of the seven ML algorithms. It can be observed that each algorithm was combined with the three attribute selection methods, and each one was executed separately, considering (Figure 4A) or excluding (Figure 4B) the “chemical family type” attribute. It is crucial to mention that the six MCC values used to construct each boxplot correspond to the six bioactivity categorization strategies (Table 1). As a result, the best and worst models are labeled using the categorization cutoffs, while the best and worst average algorithms are indicated by the red and black arrows, respectively. Based on these findings, noticeable differences among the various algorithms become evident. The algorithm with the poorest average performance is NB, regardless of whether the chemical family type is considered or not. However, the individual model with the worst MCC performance occurs when LR is used in combination with the bioactivity categorization based on a 5 µM cutoff and without utilizing an attribute selection method, yielding only a 0.25 MCC score. On the other hand, the algorithms with the best average performance are DT, when the chemical family type is not considered, and XGB, when it is considered. Both algorithms employ XGB as the attribute selection method. Regarding individual models that do not consider the chemical family type, the best is RF, using the BORUTA attribute selection method and combined with the bioactivity categorization based on 10–50 µM cutoffs (RF_BORUTA_10–50), achieving a 0.84 MCC score. The top individual models that consider the chemical family type are DT, using the BORUTA attribute selection method and combined with the bioactivity categorization based on 5–50 µM cutoffs (DT_BORUTA_5–50), achieving a 0.97 MCC score. Additionally, XGB, without an attribute selection method and combined with the bioactivity categorization based on a 5 µM cutoff (XGB_nFS_5), achieves a 0.97 MCC score.

A comparison to identify the optimal bioactivity categorization strategy for inhibitors is presented in Figure 5. In this case, each strategy is combined with the three attribute selection methods and executed separately, including or excluding the “chemical family type” attribute. It is important to note that the seven MCC values used to construct each boxplot correspond to the seven ML algorithms; hence, the best and worst models are labeled with the algorithm names, while the best and worst average strategies are indicated by the red and black arrows, respectively. Analyzing these results, the differences are not as pronounced as in the comparison of ML algorithms. However, it is observed that the best average strategy is when using the 5–50 µM cutoffs in combination with XGB as the attribute selection method. This holds true for both cases, including or excluding the “chemical family type” attribute. Since the data used to construct Figure 4 and Figure 5 (MCC scores of the models) are the same, the models with the best and worst performance coincide between both figures. Thus, as can be observed, RF_BORUTA_10–50 emerges as the best model when the “chemical family type” attribute is excluded, while DT_BORUTA_5–50 and XGB_nFS_5 are the best models when the “chemical family type” attribute is included.

The data collection process, as described in the methodology, underscores the robustness of our study. Specifically, the calculation of molecular descriptors using the rCDK package generated a comprehensive set of 290 parameters across five categories: ‘topological’, ‘electronic’, ‘constitutional’, ‘hybrid’, and ‘geometrical’. From these initial parameters, 83 were excluded due to their high variability and minimal contribution to information. The final set of 207 descriptors, along with an extra categorical attribute representing the chemical family, were meticulously explored even with Boruta and XGB as feature selection methods to ensure their relevance to physicochemical coupling with the urease binding site. This rigorous selection process aimed to enhance the predictive capabilities of our models and provide valuable insights into potential drug discovery pathways. However, there was not a clear preference for one ML algorithm over another, even with better MCC performances without feature selection (considering the 207/208 descriptors). It is also important to notate that the characteristics selected in both the supervised and unsupervised processes coincide to a large extent (Appendix A), supporting that these characteristics were a good input for the construction of classification models.

The effectiveness of tree-based methods such as RF, DT, and XGB can be attributed to several factors. Firstly, these algorithms are capable of capturing non-linear relationships and interactions between features, which are often present in complex biological datasets. Additionally, tree-based models inherently handle feature importance, allowing for the identification of key molecular descriptors contributing to bioactivity prediction. Moreover, ensemble methods like random forest and XGB further enhance predictive performance by aggregating multiple decision trees, thereby reducing overfitting and improving generalization to unseen data.

Finally, Figure 6 presents a direct comparison of the top four models, whether including the “chemical family type” attribute or not. First and foremost, it is observed that all models achieve an AUC greater than 0.93. However, the best performances are attained when the “chemical family type” of the inhibitors is considered as an attribute. As mentioned previously, the best model when excluding the “chemical family type” attribute is RF_BORUTA_10–50, which achieves an AUC of 0.9928 in this analysis. On the other hand, when the “chemical family type” attribute is considered, the top identified models are DT_BORUTA_5–50 and XGB_nFS_5, both of which achieve a perfect AUC of 1, indicating flawless classification between inhibitors and non-inhibitors. Another crucial aspect to mention regarding the algorithms is that all eight models presented in Figure 6 are tree-based methods, surpassing other models like SVM, LR, KNN, or NB. From the perspective of attribute selection methods, there seems to be no direct preference for one over the other; therefore, all of them could be viable for implementing tree-based models. As for the categorization strategies, there also appears to be no clear preference when analyzing these ROC curves. However, considering the results from Figure 5, it is inferred that the best outcomes are obtained when using the strategies with a gray zone, meaning the use of two cutoffs to categorize UIs and nUIs. The ROC curves emphasize their ability to distinguish between UIs and nUIs. Achieving high AUC values in ROC curves is crucial in drug discovery as it reflects the model’s ability to correctly classify compounds into their respective categories. High AUC values indicate strong predictive performance, suggesting that the models are capable of accurately identifying potential UIs. In a clinical context, these models could play a vital role in accelerating the drug discovery process by prioritizing compounds with a higher likelihood of urease inhibition for further experimental validation. Additionally, the biological relevance of the models’ predictive accuracy underscores their practical utility. The accurate prediction of bioactivity enables researchers to focus resources on compounds with the greatest potential for therapeutic intervention, thereby facilitating the development of novel treatments for conditions such as *Hp* infection.

## 3. Materials and Methods

### 3.1. Data Collection

A scientific literature exploration was carried out on the Web of Science (WOS) database, utilizing the search terms “urease inhibitors” AND “*Helicobacter pylori*”. The search was limited to articles published from 2010 onwards. Afterward, a categorization of inhibitors was produced by grouping them according to their chemical family. We found compounds belonging to flavonoids [27,28,29], alkaloids [30], triazole [31,32,33,34,35], thiadiazole [24,30,31,36], and coumarins [24,37,38,39,40,41] chemical families. This dataset involved an IC_50_ range from 0.009 micromolar (µM) to concentrations where a minimum inhibitory concentration for *Hp* urease enzyme was not determined, here termed non-urease inhibitors (nUIs). This is because some molecules were compounds whose inhibitory concentration was not detected in the experiments; in other words, they were molecules without inhibitory potency for *Hp* urease. Therefore, they did not have a numerical value associated with the response variable. Thus, considering the biological criteria and empirical values of non-inhibitory compounds reported in the literature, an arbitrary IC_50_ value of 1 mM was assigned to all these molecules. Lastly, those compounds whose inhibition measurement was not carried out by calculating the half-maximal inhibitory concentration (IC_50_) were discarded. The UIs and nUIs previously collected were drawn utilizing the 2D Sketcher tool from the Maestro Schrodinger suite [56], which was also used to add their corresponding valences. To assign protonation states, the Epik tool [57] was utilized at a pH of 7.2, which is the standard pH at which biological assays are typically carried out in urease. Finally, each of the 667 molecules were converted to an SDF format for subsequent analysis.

### 3.2. Characterization and Preprocessing

The calculation of molecular descriptors was performed using the rCDK package version 3.6.0. [58] from the Chemistry Development Kit library in the R programming environment [59]. All available categories of descriptors in this library (“topological”, “electronic”, “constitutional”, “hybrid”, and “geometrical”) were calculated, generating a total of 290 descriptors (Appendix A). Subsequently, data processing was carried out to discard any variable with minimal information contribution and/or to impute missing data in specific variables. To do this, firstly, a criterion was generated to exclude attributes that had 80% or more information loss. As a result of this, 286 descriptors remained, and the excluded variables were Wgamma1.unity, Wgamma2.unity, Wgamma3.unity, and WG.unity. Then, using the multiple imputation by chained equations method [60] with 3 iterations and 3 imputations, missing data were completed for the variables Weta1.unity and WD.unity. Next, variables with variance close to zero were excluded using the nearZeroVar function of the Caret package version 6.0-94 [61], resulting in a total of 207 molecular descriptors. It is important to mention that, in addition to the 207 descriptors, an extra categorical attribute corresponding to the type of chemical family previously recorded was considered. This attribute had significant relevance for the subsequent stages of the study as it may or may not have been included in the models according to the strategy used.

### 3.3. Exploratory Data Analysis

The dataset comprised 677 examples (compounds), 207 numerical variables (molecular descriptors), and 1 categorial variable (family type), and the response variable (IC_50_ in µM) was subjected to an EDA with the double aim of (1) detecting the existence of a correlation between descriptors studied and (2) identifying whether all descriptors or a subset of them enabled a clear differentiation between the UI and nUI classes in the response variable. Furthermore, IC_50_ is not a linear measure, and hence, it does not allow the adequate separation of the classes. We transformed the variable response into pIC_50_, the negative logarithm of IC_50_. This transformation is also commonly applied in statistical contexts to positive quantities to symmetrize data. On the other hand, and as mentioned before, this work aimed to predict UIs through binary classification models. For this purpose, we assigned a cutoff in the response variable to maximize the separation between the UI and nUI classes, where UIs had an IC_50_ ≤ 5 µM (pIC_50_ ≤ 5.30) and nUIs had an IC_50_ ≥ 50 µM (pIC_50_ ≥ 4.30). For the 207 numerical descriptors, we analyzed the correlation between variables through a heatmap of the correlation matrix by using the corrplot package in R. We considered variables with a correlation magnitude (regardless of the sign) of >0.4. The variables exhibiting the highest correlations with the response variable were considered the most promising candidates for constructing predictive models. This stemmed from the fact that alterations in these highly correlated variables tend to correspond with shifts in the response variable. Therefore, the correlated features were used (1) as input to analyze the central tendency (median) and variability (interquartile range) measures in UI and nUI classes with a statistical Wilcoxon rank sum test and (2) to separate both classes through a principal component analysis (PCA) using the built-in R functions prcomp(). Additionally, we visualized the chemical space of the dataset by using the Python library ChemPlot [62,63] through a t-distributed stochastic neighbor embedding (t-SNE) analysis with 1000 iterations and perplexity = 30. t-SNE is a non-linear dimensionality reduction technique that is particularly effective at preserving the local structure of the data. Here, we presented the dimensionality of UIs, nUIs, and those molecules that did not fit into the predefined classes, here called the gray zone.

### 3.4. Strategies for Bioactivity Categorization (Data Splitting)

In the previous EDA, we employed a cutoff to separate and categorize both classes. Particularly, the t-SNE analysis indicated that the chosen cutoff (5–50 µM) effectively separated the classes. However, the presence of gray points representing the gray zone highlighted the need for deeper exploration. These data points, with their ambiguous bioactivity, required further scrutiny to optimize the class separation while minimizing data loss. In this sense, in the ML scheme, more than one cutoff concentration was used for this bioactivity categorization, giving rise to different classification tasks and at the same time different strategies to predict UIs. The details of the cutoffs used, and the proposed prediction strategies, are presented in Table 1. Six strategies were planned, three based on 1 cutoff and three based on 2 cutoffs. Basically, when there were 2 cutoffs, compounds with IC_50_ values greater than cutoff 1 and lower than cutoff 2 (compounds in the gray zone) were excluded from the training and testing of the models. Instead, in the case of employing a single cutoff, no compound was excluded within the ambiguous gray zone, thus defining compounds under the cutoff as UIs and those surpassing it as nUIs. The quantities of UIs and nUIs for each strategy are presented in Table 2 and Table 3. It is important to note that the use of either IC_50_ or pIC_50_ is irrelevant for ML models as they can naturally model non-linearities through various non-linear transformations during preprocessing.

### 3.5. Training and Testing of Inhibitory Classification Models

In the six proposed strategies, the data were distributed between training and testing in an 80:20 ratio (Table 4). Seven supervised ML algorithms were used and compared to train the models: RF, SVM, DT, XGB, KNN, NB, and LR. In addition, Boruta [64] and XGB [65] were used separately as feature selection methods during training to compare them with the models built using all the attributes (nFS). Boruta utilizes a random forest-based approach to identify relevant features by comparing their importance with randomized counterparts. On the other hand, XGB assesses feature importance by training multiple decision trees and evaluating their frequency of use in decision-making processes. Repeated cross-validation (10 folds and 5 repetitions) was used to train and validate the models. Furthermore, SMOTE [66] was applied to balance the classes in each training step. Moreover, the hyperparameters of each algorithm were optimized during cross-validation using a grid search method. The models were both trained and tested with consideration for the chemical family of the compounds within the dataset and without taking this parameter into account. All ML algorithms, feature selection methods, SMOTE, and cross-validations were executed in R using the Caret package functions. Finally, to evaluate the models, the Matthews correlation coefficient (MCC) and the area under the ROC curves (AUC-ROC) were calculated.

## 4. Conclusions

In conclusion, this study provides valuable insights into the prediction of urease inhibitors using cheminformatics and ML approaches. Through a comprehensive methodology and rigorous analysis, several key conclusions can be drawn:Algorithm Preference: The study recommends favoring tree-based methods, including random forest (RF), decision tree (DT), and eXtreme Gradient Boosting (XGB), over other algorithms like k-nearest neighbor (KNN), support vector machine (SVM), naive Bayes (NB), or logistic regression (LR) for inhibitor classification.Attribute Selection Influence: While attribute selection methods could potentially improve model performance, their influence varies based on the ML algorithm chosen. There is not a clear preference for one method over another, suggesting their implementation should be algorithm-specific.Effective Categorization Strategies: The exploratory data analysis and ML analysis recommend employing strategies that involve a gray zone, utilizing two cutoffs for categorizing urease UIs and nUIs. These strategies tend to yield better model performance, offering improved accuracy in classification tasks, reaching almost 10 percent over one-cutoff strategies in our models. By delineating these boundaries, we can effectively train our models to distinguish between active and inactive compounds, thus enhancing the accuracy of our predictions. Moreover, understanding the implications of these cutoffs is critical for optimizing model performance. Nevertheless, strategies with a gray zone can lead to better performance, and it is crucial to consider the biological implications. Expanding the gray zone for categorization may result in the loss of important information about inhibitors.Consideration of Chemical Family: The inclusion of the chemical family attribute significantly enhances the classification models. However, obtaining this attribute might require manual annotation or inspection as automatic extraction from databases like ChemBL might not be straightforward. Despite the effort required, incorporating this attribute contributes to the models’ effectiveness.State of Art in Urease Inhibitors: To the best of our knowledge, in the context of drug discovery targeting *Hp* infection through urease inhibition, our study stands out as the most comprehensive and systematic evaluation of optimal conditions for developing predictive models of bioactivity for potential inhibitor candidates. By rigorously testing various attribute selection methods, machine learning algorithms, and bioactivity categorization strategies, we provide a robust framework that could significantly accelerate the identification and development of novel urease inhibitors. The elucidation of the structure–activity relationship (SAR) is crucial for rational drug design as it provides valuable information about how changes in the chemical structure of compounds affect their biological activity. Our investigation contributes to this understanding by identifying molecular features that correlate with UIs. By analyzing these relationships, researchers can gain insights into the chemical properties that are essentials for designing potent UIs.Practical Significance for Drug Design: Our approach serves as a practical guide applicable not only to urease but also to other proteins in drug design, potentially impacting the field with its systematic methodology and comprehensive evaluation. This intersection between computational modeling and biological relevance highlights the significance of our findings in advancing both drug discovery efforts and our understanding of urease inhibition mechanisms. Developing predictive models that accurately classify compounds based on their inhibitory activity against any relevant clinical target, as demonstrated in our study, enables the efficient screening of large compound libraries to identify promising drug candidates. This can significantly accelerate the drug discovery process by prioritizing compounds with the highest likelihood of exhibiting inhibitory activity.

In general, this study illustrates the effectiveness of combining seven ML algorithms, three attribute selection methods, and six different strategies for categorizing inhibitory activity to enhance the prediction of urease inhibitors. The provided recommendations offer practical guidance for researchers aiming to develop effective classification models for similar biochemical systems.

## Figures and Tables

**Figure 1 ijms-25-04303-f001:**
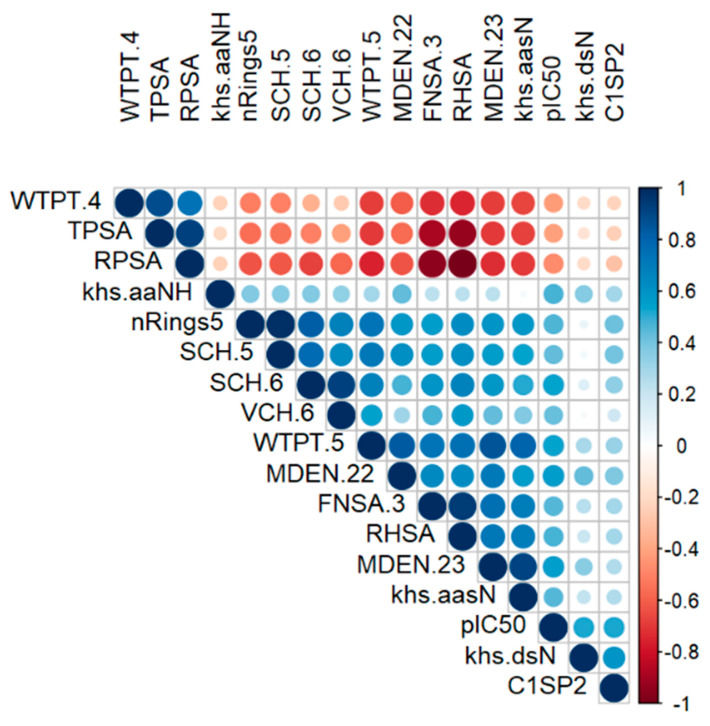
Heatmap of the correlation matrix. The heatmap was plotted by considering the negative logarithm of response variable (pIC50) and the most strongly correlated features (ρ > 0.4). The size of the circle, as well as the color, reflects the intensity of the correlation of the two variables found at the intersection of the matrix. The blue color reflects positive correlations, while the red color reflects negative correlations. Finally, white reflects an absence of correlation.

**Figure 2 ijms-25-04303-f002:**
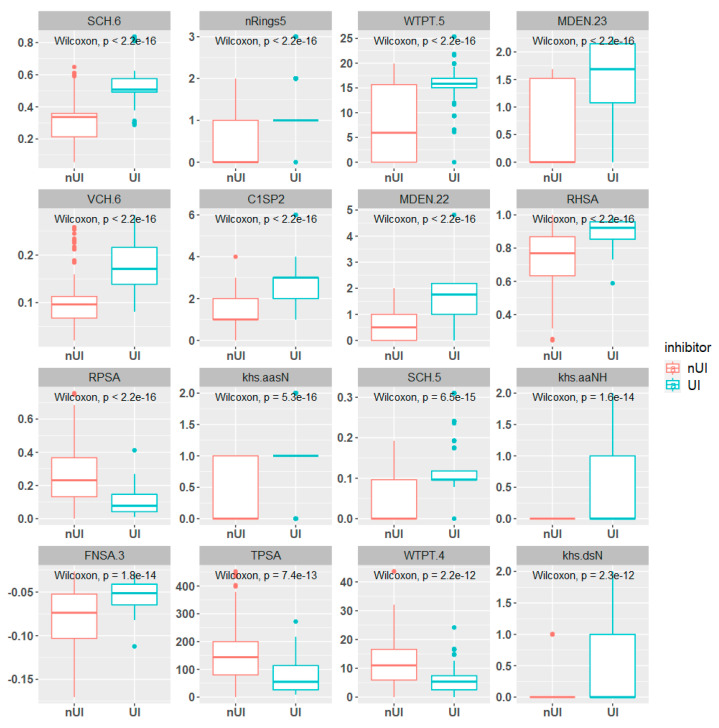
Boxplots for the 16 most relevant features. Each feature plotted was separated according to the response variable using the double cutoff 5 µM and 50 µM. The *p*-value was computed with a statistical Wilcoxon rank sum test.

**Figure 3 ijms-25-04303-f003:**
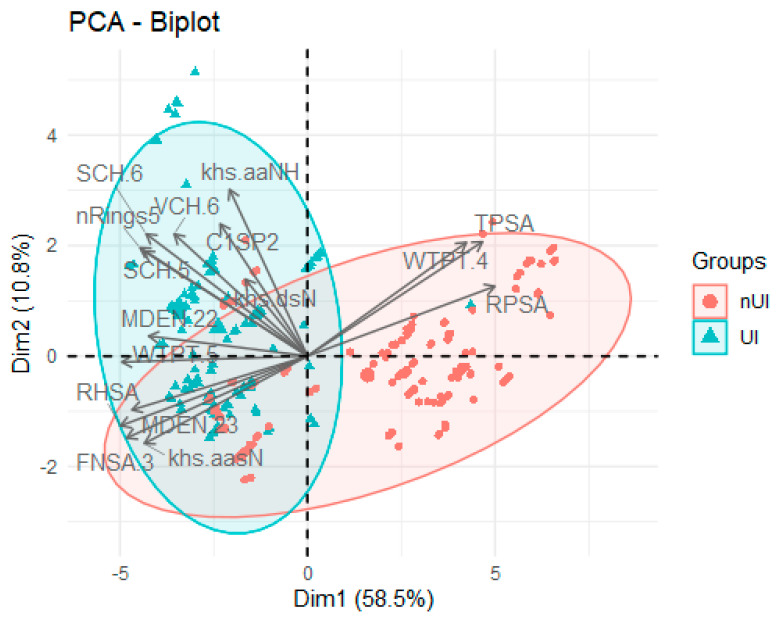
Two-dimensional representation of the data and relevant features using PCA where each figure represents a distinct compound categorized as a urease inhibitor (UI) indicated in cyan, encompassing molecules exhibiting an IC_50_ ≤ 5 µM. Similarly, non-urease inhibitors (nUIs) are denoted in red, encompassing molecules with an IC_50_ ≥ 50 µM. Additionally are showed the features contributing to each class.

**Figure 4 ijms-25-04303-f004:**
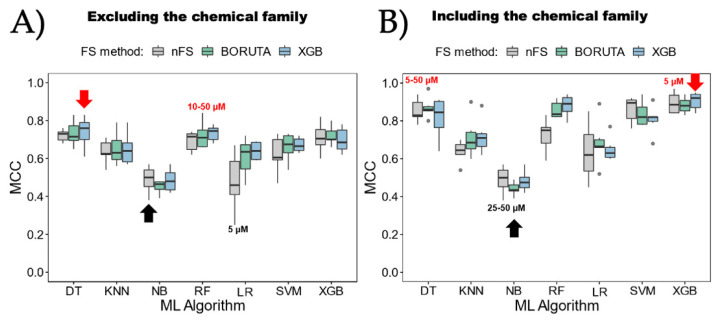
Comparison of ML algorithms. Each algorithm was compared through Matthews correlation coefficient (MCC) (**A**) excluding the chemical family and (**B**) including it. Black and red labels indicate lowest and highest MCC values individually per strategy for bioactive characterization. Meanwhile, black and red arrows show the lowest and highest MCC values as averages considering all the strategies tested in each algorithm. nFS: non-feature selection; DT: decision tree; KNN: k-nearest neighbor; RF: random forest; LR: logistic regression; SVM: support vector machine; XGB: eXtreme Gradient Boosting. The points at the ends of the boxplots show the outliers in each comparison. Calculated from those values that are below: Q1 − 1.5 ∗ IQR or above Q3 + 1.5 ∗ IQR. IQR being the Interquartile Range.

**Figure 5 ijms-25-04303-f005:**
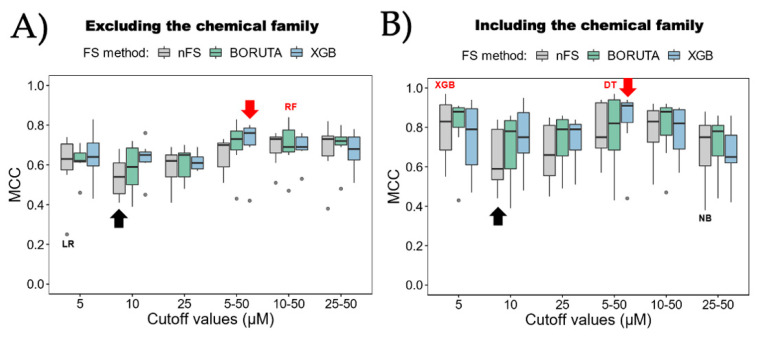
Comparison of strategies for bioactivity categorization. Each strategy was compared through Matthews correlation coefficient (MCC) (**A**) excluding the chemical family and (**B**) including it. Black and red labels indicate lowest and highest MCC values individually per ML algorithm. Meanwhile, black and red arrows show the lowest and highest MCC values as averages considering all the ML algorithm tested in that strategy. nFS: non-feature selection. The points at the ends of the boxplots show the outliers in each comparison. Calculated from those values that are below: Q1 − 1.5 ∗ IQR or above Q3 + 1.5 ∗ IQR. IQR being the Interquartile Range.

**Figure 6 ijms-25-04303-f006:**
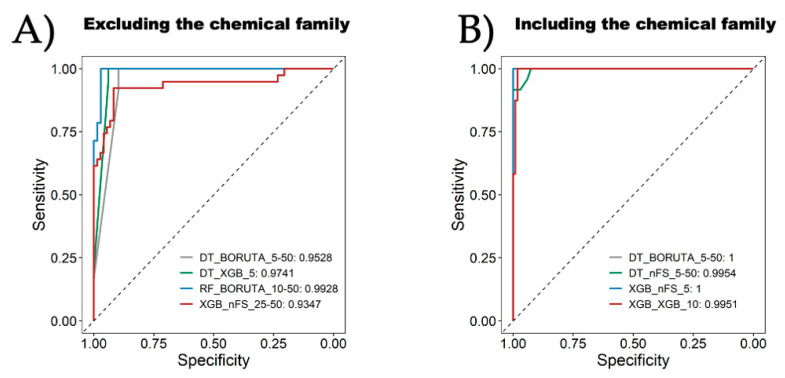
Comparison of the best four models (**A**) excluding the chemical family and (**B**) including it. The legend shows the algorithm followed by the feature selection and the strategy for bioactive characterization associated with the area under the curve (AUC) for each model.

**Table 1 ijms-25-04303-t001:** Proposed prediction strategies based on the IC_50_ values for bioactive characterization.

One-Cutoff Strategies	Two-Cutoff Strategies
UIs < 5 µM < nUIs	UIs < 5 µM < gray zone < 50 µM < nUIs
UIs < 10 µM < nUIs	UIs < 10 µM < gray zone < 50 µM < nUIs
UIs < 25 µM < nUIs	UIs < 25 µM < gray zone < 50 µM < nUIs

**Table 2 ijms-25-04303-t002:** UIs and nUIs for 1-cutoff strategies for bioactive characterization.

	IC_50_: 5 µM		IC_50_: 10 µM		IC_50_: 25 µM	
	N°	%	N°	%	N°	%
UIs	119	18	145	22	221	33
nUIs	558	82	532	78	456	67

**Table 3 ijms-25-04303-t003:** UIs and nUIs for 2-cutoff strategies for bioactive characterization.

	IC_50_: 5 µM and 50 µM		IC_50_: 10 µM and 50 µM		IC_50_: 25 µM and 50 µM	
	N°	%	N°	%	N°	%
UIs	119	26	145	30	221	39
nUIs	341	74	341	70	341	61

**Table 4 ijms-25-04303-t004:** Data distribution between training and testing per strategy.

1-Cutoff Strategies			2-Cutoff Strategies		
IC_50_	Training	Testing	IC_50_	Training	Testing
IC_50_: 5 µM	542	135	IC_50_: 5 µM and 50 µM	368	92
IC_50_: 10 µM	542	135	IC_50_: 10 µM and 50 µM	389	97
IC_50_: 25 µM	542	135	IC_50_: 25 µM and 50 µM	450	112

## Data Availability

Data are contained within the article and Appendix A.

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
