# Peer review of "Machine Learning-Driven Classification of Urease Inhibitors Leveraging Physicochemical Properties as Effective Filter Criteria"

_ijms, 2024, doi:10.3390/ijms25084303_

Round 1

Reviewer 1 Report

Comments and Suggestions for Authors

General Assessment:

This manuscript presents a comprehensive study on the prediction of urease inhibitors using machine learning (ML) models, emphasizing the utility of physicochemical properties for enhancing predictive accuracy. The authors meticulously compiled a dataset of urease inhibitors, calculated their physicochemical properties, and applied various ML algorithms to develop classification models. The study's strength lies in its detailed methodological approach, encompassing data collection, preprocessing, exploratory data analysis, and the rigorous evaluation of ML models across multiple categorization strategies. The work is significant for its potential to guide future drug discovery efforts in identifying potent urease inhibitors, particularly against Helicobacter pylori, contributing valuable insights into the computational prediction of bioactive compounds.

Major Comments:

1.      Originality and Novelty: The manuscript's application of 252 classification models combining seven ML algorithms, three attribute selection methods, and six different categorization strategies for bioactivity (page 7, Lines 209-218) is commendable for its thoroughness. Comparing these strategies to existing models in the literature could highlight the novelty more vividly. For example, the work's differentiation through its extensive and systematic approach to model construction and evaluation sets a new benchmark in the field.

2.      Methodology:

a.      The data collection process, involving an extensive literature search and characterization based on physicochemical properties (page 3, Lines 102-107), provides a robust foundation for the study. Clarification on the selection criteria for these properties and their expected impact on the models' predictive capabilities would add valuable insights into the study's methodological rigor.

b.      The introduction of a "gray-zone" categorization (page 5, Lines 184-189) for compounds with ambiguous bioactivity is a novel aspect of the methodology. Further elucidation on how this categorization was determined and its implications for model accuracy and drug discovery would be beneficial.

3.      Results and Discussion:

a.      The finding that tree-based methods (Random Forest, Decision Tree, and XGBoost) showed superior performance (page 8, Lines 229-234) is a critical result that could shape future computational approaches in the field. Discussing the potential reasons behind the effectiveness of these algorithms and their implications for the computational prediction of bioactive compounds could provide a deeper understanding of the study's impact.

b.      The manuscript briefly touches upon the significant enhancement in model accuracy when incorporating the "chemical family type" attribute (page 7, Lines 209-214). Expanding on how this finding aligns or contrasts with previous studies and its potential applications in predictive modeling would enrich the discussion.

Minor Comments:

1.      Figures:

a.       Figure 1 (Heatmap of the Correlation Matrix), This figure effectively visualizes the strength of correlation between various physicochemical properties and the bioactivity indicator pIC50. Notably, it highlights attributes with a correlation magnitude greater than 0.4, providing a clear visual distinction between strongly and weakly related variables. Suggesting an expansion on this figure's discussion in the text to explain how these correlations influenced the subsequent selection of features for the ML models could provide readers with deeper insights into the predictive modeling process.

b.      Figure 3 (Two-Dimensional Representation Using PCA), This figure provides a PCA-based visual differentiation between UIs and nUIs, with different features contributing to each class. Expanding on how PCA helped in understanding the feature space and the clustering of inhibitors versus non-inhibitors could offer valuable context. Moreover, a discussion on the overlap between classes in the PCA space and its impact on model performance would provide a deeper analysis.

c.       Figure 6 (ROC Curves of Top 4 Models), Presenting the ROC curves for the top models highlights their discriminative ability between UIs and nUIs. Discussing the significance of achieving high AUC values in the context of drug discovery and the potential clinical implications of these models could add a valuable dimension to the manuscript. Furthermore, considering the biological relevance of these models' predictive accuracy could offer insights into their practical utility.

Comments on the Quality of English Language

 Grammar and Style: While the manuscript is generally well-written, attention to detail in proofreading for typographical errors (e.g., "Decision Tree (DT), and eXtreme Gradient Boosting (XGB) over other algorithms like k-Nearest Neighbor (KNN), Support Vector Machine (SVM), Naive Bayes (NB), or Logistic Regression (RL)" - Logistic Regression should be abbreviated as LR, not RL) would polish the manuscript for publication.

Author Response

Dataset: https://www.dropbox.com/scl/fi/oe4p158byw6m82rrscgzn/Dataset.xlsx?rlkey=xa5wx2pvp24tzz660mr4q2be7&dl=0

Features_selected: https://www.dropbox.com/scl/fi/elz6auk9fgxmzuwzny83l/Features_selected.xlsx?rlkey=cg689ykzko341rb017u8fqp07&dl=0

Manuscript reviewed: https://www.dropbox.com/scl/fi/ux0zjq7jvhn05afeboyxb/Manuscript_IJMS.docx?rlkey=ik44ukbycvppa7a0a1ej88792&dl=0

Supplementary information: https://www.dropbox.com/scl/fi/s3tkc2px3met5eg0xwa25/supplementary_information_IJMS.docx?rlkey=uktz2zsimaszrr2h80na548di&dl=0

Reviewer 2 Report

Comments and Suggestions for Authors

 Comments to the Author

In this study, the authors developed the Machine Learning models to predict urease inhibitors using the physicochemcal properties. Total, 252 classification models were trained, utilizing a combination of seven ML algorithms, three attribute selection methods, and six different strategies for categorizing inhibitory activity. This manuscript could be considered publishing after revision.

Some comments and questions are as follows:

[1] Review improve the English or grammatical errors throughout the text.

[2] It will be good to the reader, if authors provide overall workflow of the study in one figure.

[3] Please elaborate on the implications of this study for understanding the structure-activity relationship of urease inhibitors and its practical significance for drug design and discovery.

[4] The dataset of compounds used in this manuscript could be contained in Supplementary material to reproduce the work.

[5] Authors should provide the diversity of data set used in the study. Diversity is an important factor in developing the robust models. Authors should use fingerprints to calculate the diversity and discuss it in manuscript. 

[6] In Figure 6. The title “not considering the chemical family” and “considering the chemical family” is misleading. Authors should use appropriate title for the figure. This should be changed in throughout the manuscript.

[7] Page 9 and Line no 305. “The calculation of molecular descriptors was performed from molecules in SDF format using the rCDK package version 3.6.0”. Please replace this line with “The calculation of molecular descriptors was performed using the rCDK package version 3.6.0.”

[8] Page no. 10 and Line no 32. The authors mention they have used 677 compounds in their study. How they have selected these compounds? Whether they survey any publicly available chemical databases like ChEMBL, PubChem, ZINC etc.?

[9] If possible, the authors also provide the consensus outcomes of all seven ML models.

Author Response

(The authors gave the same response as above.)

Round 2

Reviewer 1 Report

Comments and Suggestions for Authors

The authors have addressed reviewer feedback satisfactorily.